# Identification of Single-Nucleotide Polymorphisms in ZNF469 in a Patient with Aortoiliac Aneurysmal Disease

**Adam Wolf** [1], **Faria Khimani** [1] and **Mohanakrishnan Sathyamoorthy** [2,3,*]

1 Sathyamoorthy Lab, Texas Christian University School of Medicine, Fort Worth, TX 76129, USA; a.wolf@tcu.edu (A.W.); faria.khimani@tcu.edu (F.K.)
2 Department of Medicine, Texas Christian University School of Medicine, Fort Worth, TX 76129, USA
3 Consultants in Cardiovascular Medicine and Science—Fort Worth, 1121 5th Avenue, Suite 100, Fort Worth, TX 76104, USA
* Correspondence: m.sathyamoorthy@tcu.edu; Tel./Fax: +1-817-423-8585

**Abstract:** Thoracic aortic aneurysms and dissections often have inter-related pathologies that are increasingly recognized to have a genetic basis. A patient with a vascular history consisting of a spontaneous aorto-iliac dissection and thoracic aortic aneurysm belonged to a family with a significant self-reported history of aneurysmal disease. Suspecting a genetic component, genetic investigation was undertaken. Three variants of unknown significance were found in the ZNF469 gene, which is responsible for the production of a collagen-related zinc finger protein involved in multiple aspects of the development and regulation of major extracellular matrix components. This is the first report to associate this gene with vasculopathy, and further investigation by our group is underway to understand the role it plays in the development of aneurysmal diseases.

**Keywords:** ZNF469; aortic aneurysm; iliac aneurysm; aortic dissection; iliac arterial dissection; genetic basis of aortic disease; aortopathy

## 1. Introduction

Thoracic aortic aneurysms (TAA) can generally be defined as an ascending aorta with an axial diameter greater than 1.5 times the expected diameter secondary to the weaking of the vessel wall. This disease process is rare, even more so than abdominal aortic aneurysms, and occurs in roughly 6–10 individuals per 100,000. Symptoms of TAA are not always present but can include hoarseness, dysphagia, and chest or back pain. This pathologic state is often discovered incidentally and management is based on size, symptom profile, and underlying etiology. Feared complications of TAA include dissection and rupture, both of which are life-threatening complications that require emergent intervention. Investigation into the etiologies of TAA has become increasingly multidisciplinary and relies heavily on genetic testing. It has been reported that for ascending aortas measuring >4.0 cm, dissection occurs in 1.5% of patients and rises to 3.7% at >6.0 cm [1]. Whether a consequence of aneurysmal disease or as a standalone pathology, vascular dissections are often serious and potentially fatal, especially when a major artery is involved. Similar to aneurysmal disease, dissections of vessels develop when structural degeneration gives way to the continuous effects of systemic arterial pressure.

The etiology of TAAs can be broken into the three categories consisting of familial non-syndromic, syndromic, and spontaneous. Familial thoracic aortic aneurysmal and dissection (TAAD) diseases are rare, though approximately 30 identifiable genetic mutations are currently recognized [2]. Syndromic disorders, which include Ehlers-Danlos and Marfan syndromes, often share a phenotypic endpoint of cystic medial necrosis, characterized by decreased medial ground-substance, smooth muscle apoptosis, and increased elastic fragmentation [3]. Non-syndromic causes and risk factors center around the stiffening of the vessel, namely arteriosclerosis intertwined with effects from chronic plaque, i.e.,

atherosclerosis. This underlying pathology is further potentiated by increasing age, dyslipidemia, chronic hypertension, and tobacco smoking. These risk determinants contribute to the degeneration of aortic media components, including elastin and collagen leading to the replacement of elastic tissue by more fibrotic tissue. Other predisposing factors contributing to TAAs include the outcome of chronic infections, such as tertiary syphilis, mycoses, or may be associated with valvular pathologies such as a bicuspid aortic valve [4–6].

In this paper, we report the case of a patient with a dilated ascending aorta and bilateral spontaneous aortoiliac dissections, in whom we have identified a novel mutation in ZNF469 which we believe may define a genetic basis for her disease state.

## 2. Methods

The patient in this report underwent genetic screening of 35 genes associated with aneurysmal and dissection disease and related disorders via genomic deoxyribonucleic acid-isolated salivary sampling. Enrichment of targeted coding exon sequences was carried out by bait-capture methods using biotinylated oligonucleotide probes and subsequent polymerase chain reaction and sequencing, based on NCBI reference sequences [7].

## 3. Case Description

A 71-year-old African American female with a past medical history significant for chronic, benign hypertension well controlled on metoprolol succinate 25 mg once a day presented to our practice in 2014 for a cardiac evaluation. Her index transthoracic echocardiogram (TTE) revealed preserved left ventricle (LV) function, normal LV end-systolic and end-diastolic dimensions, and moderate-to-severe aortic insufficiency (AI). At the time her aorta measured normal in dimension. Within a year from this evaluation, her AI progressed to severe with symptoms, leading to aortic valve (AV) replacement with a bioprosthetic valve in 2015. In 2018, the patient developed symptoms that on a suboptimal transthoracic echocardiographic evaluation suggested prosthetic AVR dysfunction. For definitive characterization, she underwent a transesophageal echocardiogram (TEE) which demonstrated a mildly dilated aortic sinus of Valsalva and ascending aorta at 3.94 cm and 3.9 cm diameter, respectively, but normal prosthetic AV function. We noted the change in aortic dimensions in comparison to prior imaging and planned for serial follow-up. In 2019, she presented to her primary physician with abdominal and back pain worrisome for a vascular etiology. Serum laboratories (CBC, CMP, PT/INR) and an EKG were normal. Upon urgent referral to our program, a computed tomography angiogram of her aorta with bilateral runoff demonstrated bilateral distal aortoiliac dissections, with the right iliac dissection demonstrated in Figure 1.

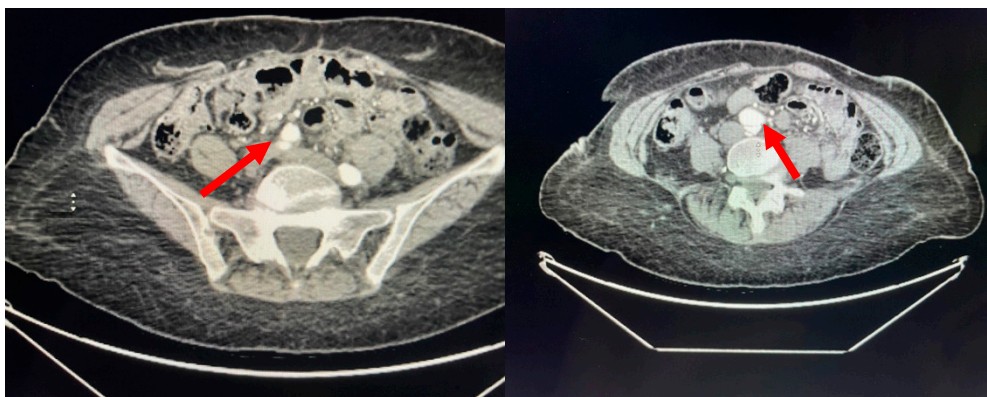

**Figure 1.** Aortic CTA with runoff. (R) Iliac dissection and false lumen indicated by arrows.

Our vascular surgical colleagues repaired these at our institution using an endovascular approach resulting in an excellent clinical outcome.

In 2021, she underwent repeat TEE to assess for prosthetic AVR dysfunction that surprisingly demonstrated dilation of the ascending aorta to 4.5 cm. It is intriguing to note the dilation pattern, as the aorta appears to literally stretch from the annulus of the seated bioprosthetic valve (Figure 2).

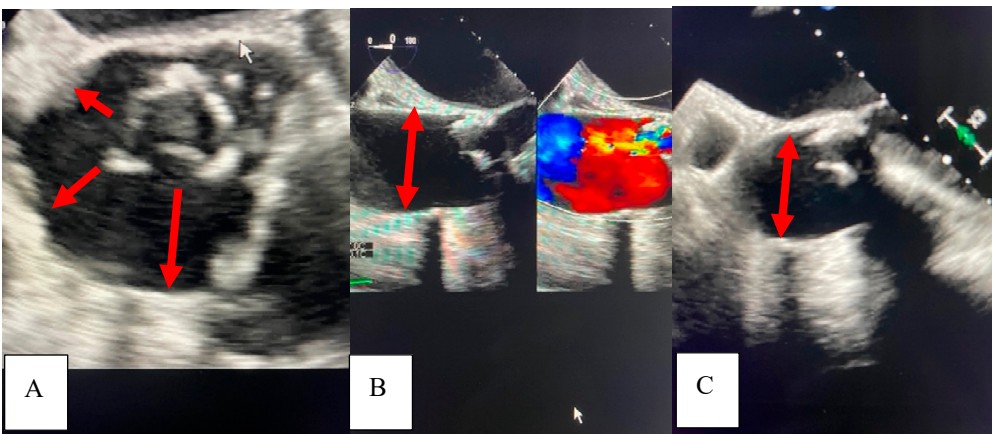

**Figure 2.** TEE demonstrating well-seated tissue bioprosthetic valve with normal 2D/Doppler function. Panel (**A**), Omniplane at 45 degrees. Panel (**B**), Omniplane at 125 degrees. Panel (**C**), Omniplane at 0 degrees. In each panel, note the enlarged root and "stretching" of the aorta away from the annulus indicated by the red arrows.

A subsequent thoracic CTA demonstrated a 4.6 cm ascending aortic aneurysm (Figure 3A), demonstrating a significant enlargement from 3.9 cm in the previous study performed on the same scanner at the same institution two years earlier in 2019 (Figure 3B).

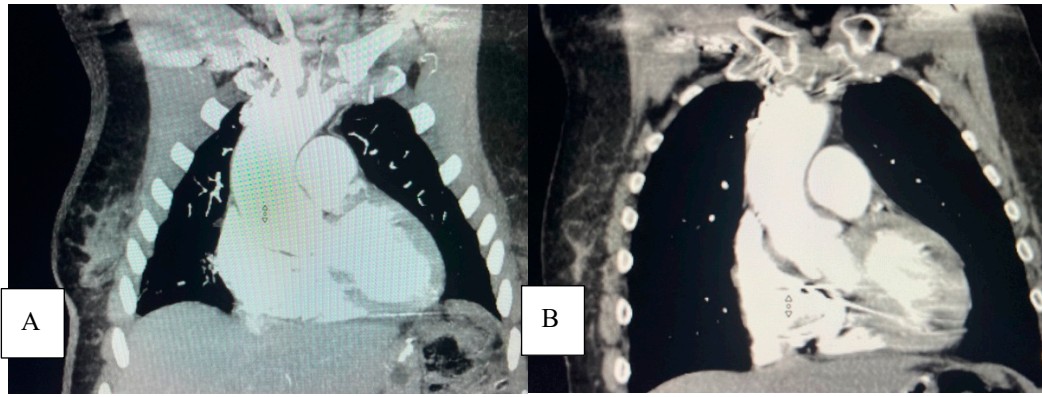

**Figure 3.** Panel (**A**), aortic CTA from January 2021 demonstrating *enlargement* of the ascending aortic aneurysm measuring 4.6 cm. Panel (**B**), aortic CTA from November 2019 demonstrating a 3.9 cm ascending aortic dimension.

After further discussion with our patient, we learned of a much more significant family history of aneurysmal disease affecting numerous family members across several generations, which allowed us to develop a family pedigree (Figure 4). A large number of her relatives are deceased and those living do not as of yet have a diagnosis or knowledge of aneurysmal diseases.

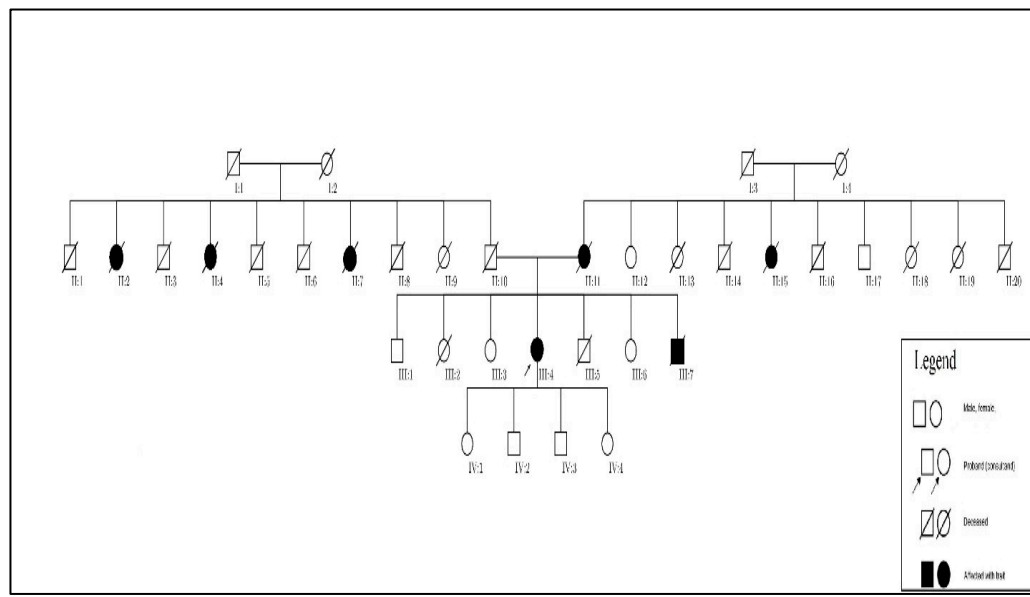

**Figure 4.** Family pedigree of aneurysmal and dissection disease. Affected members are shaded in black, indicating presumed aneurysmal or dissection disease based on patient provided history.

Given the family history, her enlarging aorta, and spontaneous subacute iliac dissections, we strongly suspected an underlying genetic basis of her disease. We performed genetic screening of 35 genes associated with TAAD diseases using the Ambry Genetics panel, which revealed three heterozygous variants of unknown significance (VUS) in the coding exon 2 of zinc finger (ZNF) protein 469 (ZNF469) gene: p.S2637T, p.G2871S, and p.Q3094R. These single-nucleotide polymorphisms (SNP) account for the replacement of serine to threonine, glycine to serine, and glutamine to arginine, respectively. The mutations exhibited in the ZNF469 VUS p.S2637T and p.G2871S produce amino acids with similar properties, with Grantham Differences of 58.00 and 56.00, respectively, while the p.Q3094R variant produces highly similar amino acid properties with a Grantham Difference of 43.00. Though all three mutations in this patient were considered tolerated in silico, these mutations provide potential for structural disruption of collagen via alterations of polarity, hydrophilicity, and secondary structure propensity.

Though the definitive clinicopathologic role of these mutations in ZNF469 is yet to be determined, our report is the first to make this association between this clinical phenotype of aorto-iliac disease and ZNF469.

## 4. Discussion

It is estimated that a contributing, identifiable genetic variant occurs in approximately 20% of TAAD cases, and, in recent years, has become increasingly important in helping characterize disease pathogenesis and risk [8]. The genetic components of the three major categories, namely the syndromic and familial non-syndromic, can be further subdivided into transforming growth factor-β vasculopathies and smooth muscle contraction vasculopathies, though there are novel genetic variants that are not represented by either category. With up to 30 genetic variants identified, the screening for such underlying factors can help predict presentation, dissection risk, and genotype–phenotype relationship in a pathology laden with variable expressivity and incomplete penetrance [2]. The identification of novel genetic variants related to aneurysmal disease will broaden screening to (1) identify patients at risk earlier in their course and/or disease state and (2) enable the development of screening paradigms to enhance clinical decisions to improve outcomes in these patients.

The zinc finger motif is one of the most common eukaryotic motifs, having been identified in a wide array of proteins, and has been shown to play an important role in vascular formation and pathology alike [9]. This diversity of function is exhibited

through its potential ability to engage in transcription, translation, and post-translational modification-related activities. The ZNF469 gene is located on chromosome 16q24.2 and is a 5 exon coding gene (Figure 5) with up to ~130 promoter and enhancer regions that produce a poorly understood C2H2 zinc finger protein present in certain collagens. As such, it is thought to be involved in the regulation of production or organization of collagen fibers [10]. There has been differing findings on the exact structure and makeup of this gene, as in 2013, Rohrbach et al. postulated that ZNF469 produces a protein product encompassing 13,279 base pairs from a single-exon gene with an internalized intron [11].

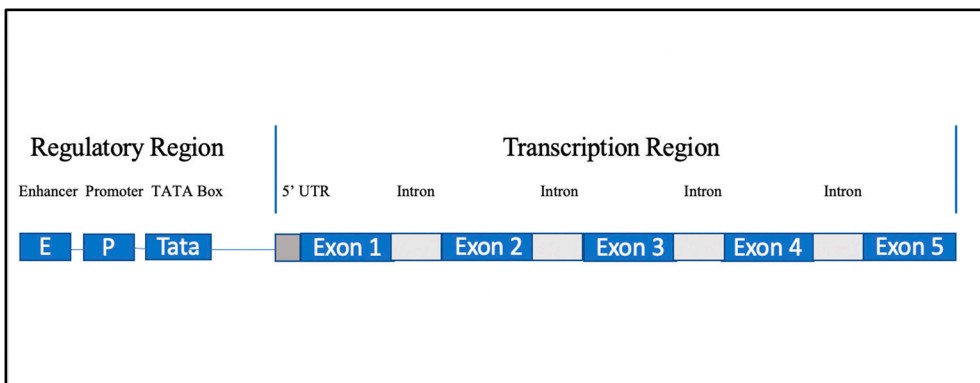

**Figure 5.** Simplified gene structure of ZNF469.

Further investigation of this gene indicated that it likely plays a role in the regulatory pathway of several ECM components. This group demonstrated that induced mutations in a dermal fibroblast ZNF469 gene lead to significant increases in Clusterin, Glypican, and Procollagen C-endopeptidase enhancer 2, and a significant reduction in Thrombospondin while immunofluorescence staining showed a disarray of Collagens I and III, fibronectin, and their receptor integrins [11]. Stanton et al. demonstrated in a murine knockout model that an induced loss of function mutation in ZNF469 leads to decreased biomechanical strength of the cornea, likely through ECM dysfunction [12].

In human beings, ZNF469 mutations are currently associated with the pathogenesis of brittle cornea syndrome, an autosomal recessive connective tissue disease characterized by extreme corneal thinning and fragility, as well as diffuse connective tissue dysfunction [11]. In addition to its role in proper ECM functioning, multiple genome-wide association studies have found the ZNF469 locus as a corneal thickness determinant [13,14]. Such disruption of major protein components supports our hypothesis that ZNF469 loss of function mutations cause vascular disruption through ECM dysfunction. There is no published data to date on gain of function transgenic or loss of function knockout animal models that guide this phenotypic characterization.

## 5. Conclusions

TAAD diseases are devastating pathologies that often result in fatal sequelae and further research into cases of seemingly non-syndromic TAADs may result in increased discoveries of novel pathologic mutations. This report is the first to make an association between TAAD and ZNF469. At the time of this report, we have initiated a familial genotyping effort among living members of her family based on the pedigree we developed from our patient's understanding of aneurysm-related disorders in her family (Figure 4). We anticipate that the findings from this familial investigation will build on the relationship between ZNF469 and arterial vasculopathy.

**Author Contributions:** A.W. authored large sections of this manuscript, conducted patient interviews, and created figures. F.K. contributed to manuscript review and revisions. M.S. made the phenotype genotype association in this patient, supervised genetic testing and counseling, developed the overall hypothesis, and was responsible for the editorial review of the entire manuscript. All authors have read and agreed to the published version of the manuscript.

**Funding:** This research received no external funding. However, the authors acknowledge the generous support of the Sathyamoorthy Laboratory from the Potishman Foundation, Fort Worth, Texas.

**Institutional Review Board Statement:** The patient in this paper provided fully informed written consent to be the subject of this report. Ethical review and approval were waived as this is a single subject case report, in accordance with IRB policy.

**Informed Consent Statement:** Written informed consent has been obtained from our patient to report all findings, including archived images related to her case in this paper.

**Data Availability Statement:** Not applicable.

**Conflicts of Interest:** The authors declare no conflict of interest.

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
