# Peer review of "Identification of Single-Nucleotide Polymorphisms in ZNF469 in a Patient with Aortoiliac Aneurysmal Disease"

_cardiogenetics, doi:10.3390/cardiogenetics12030020_

Round 1

Reviewer 1 Report

Dear Authors,

I found the article submitted for review exciting. It offers new light in cases of aortoiliac aneurysms.

I believe that the authors have correctly presented this problem and applied the correct methodology.

The summary is correct and gives a good overview of the article's content.
Introduction - this is a good introduction to the topic of the article. My only remark concerns Figure 1.
While the schematic idea itself could be pretty interesting, the implementation is poor. Little can be understood from Figure 1. It is difficult to say what the blue "balls" and the black threads in part b) mean. Furthermore, the abbreviation (part b) of the ECM was not clarified. I think a legend would be helpful here instead of the description blocks, or the authors could put the arrows with markings and show the differences between A and B - I believe they will be more valuable here.

A case report of a 71-year-old patient was correctly prepared.

However, the results of the imaging studies would have been helpful to have a better view of this patient's history if it is available. I am also deeply interested in family history. Is it possible to extend the description of the family history?
Is this patient's medical history available before 2014? The next question is: did the patient suffer from any chronic medical condition (diabetes and others), hypertension, chronic kidney disease, or taking any medications regularly? And what about her laboratory parameters: lipids, glucose, hsCRP (if available) - adding this data, I think, would help give a better picture of the patient and be helpful for the readers of the article.

A more detailed methodology (how the SNP was assessed) would be helpful - maybe even a separate chapter to not combine the methods with the case study. It doesn't have to be a long description. I recommend that you separate these two parts from each other.
I remember this is a case study, not a regular article.

The discussion -  is correct - I have no objections here.

The conclusions are correct.

Kind regards

Author Response

We deeply appreciate these excellent comments and will revise the manuscript to address all of these important points.  Of note, we will include the family pedigree and discussion centered around this.

Reviewer 2 Report

In the case report "Identification of Single Nucleotide Polymorphisms in ZNF469 in a Patient with Aortoiliac Aneurysmal Disease" the authors describe for the first timer the association between  ZNF469 gene and vasculopathy in a patient with Aortoiliac Aneurysmal Disease. 

The paper is well written and organized and of interest for the readers of cardiogenetics 

Minor points 

- Please add the family tree of the patient

- Please add the echocardiographic (both TTE and TEE) images of the patient

- Please add the  CT images of the patient 

Author Response

We completely agree with these excellent suggestions and will revise to include as many of these as possible.  Some older images may not be retrievable due to archival challenges, but every effort will be made, especially with more recent imaging.